# Soft disorder modulates the assembly path of protein complexes

**Beatriz Seoane**[1]*, **Alessandra Carbone**[2]*

**1** Departamento de Física Teórica, Universidad Complutense de Madrid, Madrid, Spain, **2** Sorbonne Université, CNRS, IBPS, Laboratoire de Biologie Computationnelle et Quantitative - UMR 7238, Paris, France

* bseoane@ucm.es (BS); alessandra.carbone@lip6.fr (AC)

**Data Availability Statement:** All the data used for these analyses (hierarchies of interaction and soft disorder) is available at the website http://www.lcqb.upmc.fr/softdisorder-assembly/.

**Funding:** This work was supported by the Comunidad de Madrid and the Complutense

## Abstract

The relationship between interactions, flexibility and disorder in proteins has been explored from many angles over the years: folding upon binding, flexibility of the core relative to the periphery, entropy changes, etc. In this work, we provide statistical evidence for the involvement of highly mobile and disordered regions in complex assembly. We ordered the entire set of X-ray crystallographic structures in the Protein Data Bank into hierarchies of progressive interactions involving identical or very similar protein chains, yielding 40205 hierarchies of protein complexes with increasing numbers of partners. We then examine them as proxies for the assembly pathways. Using this database, we show that upon oligomerisation, the new interfaces tend to be observed at residues that were characterised as softly disordered (flexible, amorphous or missing residues) in the complexes preceding them in the hierarchy. We also rule out the possibility that this correlation is just a surface effect by restricting the analysis to residues on the surface of the complexes. Interestingly, we find that the location of soft disordered residues in the sequence changes as the number of partners increases. Our results show that there is a general mechanism for protein assembly that involves soft disorder and modulates the way protein complexes are assembled. This work highlights the difficulty of predicting the structure of large protein complexes from sequence and emphasises the importance of linking predictors of soft disorder to the next generation of predictors of complex structure. Finally, we investigate the relationship between the Alphafold2's confidence metric pLDDT for structure prediction in unbound versus bound structures, and soft disorder. We show a strong correlation between Alphafold2 low confidence residues and the union of all regions of soft disorder observed in the hierarchy. This paves the way for using the pLDDT metric as a proxy for predicting interfaces and assembly paths.

## Author summary

Both flexibility and intrinsic disorder are used as regulatory mechanisms in proteins. They can alter the spatial positions of important recognition sites, and increased mobility appears to facilitate ligand binding through conformational selection. In this work, we show statistical evidence that soft disorder is directly involved in the process of protein assembly and that migration of soft disorder after binding gives rise to new or altered

University of Madrid through the Atracción de Talento program (2019-T1/TIC-12776 to BS); the Banco Santander and the Complutense University (PR44/21-29937 to BS); the Ministerio de Ciencia e Innovación, la Agencia de Investigación (Spain) and the European Regional Development Fund PID2021-125506NA-I00 to BS); the French Agency for Research on AIDS and Viral Hepatitis (ANRS–AAP2021CSS12-D21342 to AC). The funders had no role in study design, data collection and analysis, decision to publish, or preparation of the manuscript.

**Competing interests:** The authors have declared that no competing interests exist.

functions in the protein complex. Given the impressive progress that AlphaFold2 has made in protein structure prediction in recent years, this work highlights the importance of also correctly predicting conformational heterogeneity, mobility and intrinsic disorder in order to access the full functional repertoire and interaction network of a given protein.

## Introduction

Structural biology is undergoing a complete revolution thanks to modern deep-learning algorithms. Among them, Alphafold2 (AF2) is able to predict the three-dimensional structure of protein amino acid (AA) sequences with atomic accuracy for the first time in history [1, 2]. Moreover, the power of these tools goes beyond individual proteins: protein complexes are now accessible from sequence [3–7]. These results will certainly drastically increase the amount of available structures of proteins and protein complexes. Not for nothing, thanks to fast and cheap modern genome sequencing techniques, the number of candidates for viable protein sequences is several orders of magnitude larger than the number of experimentally validated structures. However, all this mainly concerns the well-structured proteins. What then happens to all those proteins that are known to be fully or partially disordered under physiological conditions [8, 9]? Currently, AF2 predictions leave these intrinsically disordered proteins (IDPs) and protein regions (IDPRs) unstructured or predict structures for regions that undergo a transition from disorder-to-order upon interaction with some partners [4, 10].

After decades of study, it now seems clear that IDPs/IDPRs play an important role in promoting and tuning protein interactions with other partners, anticipating that knowledge of disorder will be crucial for automatically predicting the structure of new or large protein complexes. Indeed, compared to well-structured proteins, IDPs and IDPRs have a large capacity to bind to multiple partners [11–14]. For example, IDPs/IDPRs are known to be rich in molecular recognition features or motifs used for protein-protein interactions [15–17]. In addition, many are observed to undergo a transition from disorder to order to interact with other partners [18–20], or even fold into alternative structures depending on which partner is involved in the interaction, often resulting in unrelated or even opposite protein functions [21]. In summary, IDPs and IDPRs are likely to promote disorder-based mechanisms that could determine the assembly of protein complexes [22].

Protein flexibility appears to play a double role in complex assembly and functional regulation. Recent work has highlighted the use of highly mobile regions to select conformations, tune different protein functions and promote new interactions [23, 24]. More complex interaction mechanisms have been reported in which new highly mobile regions are generated in distant regions following binding. Furthermore, this allosteric response appears to be associated with the appearance of a new or altered function in the complex and the creation of new interfaces [23, 24].

Despite the ubiquity of IDPs/IDRPs and flexibility, their role in protein interaction networks is usually discussed only in terms of the phenomena observed in a small number of proteins. Although several predictors of disordered region binding have been developed [18–20], there is very little statistical evidence revealing the general role of structural disorder and flexibility in complex assembly. Recently, we have carried out a large-scale analysis of disorder in known structures that provides the statistical basis mentioned above, namely the statistical correlation between the location of disorder and interfaces. Evidence for the correlation between the location of binding and disorder regions was examined in [25] using the full set of experimental X-ray structures stored in the Protein Data Bank (PDB). The results clearly show that

after cross-analysis of all alternative structures containing a particular (or very similar) protein sequence, interfaces occur with a statistically significant preference in those AAs characterised as disordered, typically in different PDB structures. However, it has also been shown that for stronger correlation it is necessary to extend the definition of structural disorder from the standard *missing residues* in the PDB structure to all those residues that are poorly resolved in the experiments, i.e. hot loops, flexible or even spatially amorphous (but rigid) regions of the protein [26, 27]. This softened version of disorder was termed *soft disorder*. These results suggest that new interfaces tend to settle into the floppy parts of a protein, and point to the idea that soft disorder may actually mediate the order in which protein complexes are assembled. With this intuition in mind, several examples of progressive assembly were discussed in [25], describing an interaction mechanism mediated by disorder that is very similar to some hypothetical mechanisms previously proposed in the IDP literature [22], and the general picture emerging from recent works on flexibility [24]. In this work, we go beyond previous studies and provide statistical evidence for the central role of soft disorder in the progressive assembly of protein complexes. We carefully exclude the possibility that this role is merely a surface effect of the protein. Upon oligomerisation, the location of soft disorder regions may change place in the protein structure, and we observe that this new location correlates with the regions where we observe new binding at higher levels of oligomerisation. A similar result is observed in unbound structures that seem to carry information about all alternative new binding regions, although it might be hard to distinguish the signals from the different binding interfaces from the sequence. On the one hand, this work highlights the importance of correctly predicting the flexible/disordered regions of a given protein complex in order to know where new partners can be accommodated. On the other hand, however, it also shows that the soft disorder depends on the structure of the intermediate complexes. In other words, selectively predicting the position of the interface region (IR) in a given complex (among all possible interactions of a given protein) based on sequences is a difficult problem.

We structure the paper as follows. First, we introduce the notion of soft disorder and explain the construction of directed graphs describing hierarchies of progressive complex assembly. We then test the notion of soft disorder as an interface predictor for new interactions in the hierarchy. We then show that the correlation between soft disorder and interfaces persists when trivial correlations between the two, such as surface residues with a higher b-factor or interface regions with a lower b-factor, are removed. We conclude with a comparative analysis of soft disorder and low confidence of AF2 in some protein complexes.

## Results

### Definition of soft disorder

In this work, we restrict our analysis of disorder to residues that are poorly resolved in a PDB X-ray structure for various reasons, i.e., floppy, highly flexible, fluctuating, or to the amorphous rigid regions of proteins. These residues can be identified by their anomalously high B-factor (or temperature) factor [27, 28], or by the missing residues in PDB structures. We also analyse the disorder of a protein chain across different alternative PDB crystals of the chain that exhibit the same interaction complexity (we will specify this idea later). As in [25], we use the union of high b-factor residues and missing residues across crystals to define a softer notion of structural disorder for a chain, which we call *soft disorder*. We would like to emphasise that a high B-factor is typically associated in the literature only with protein flexibility, but mobility is not the only reason that affects the quality of X-ray crystallography experiments. Rigid but amorphous regions (in the sense that they are not reproducible in different unit cells) also produce high B-factors or "missing" regions [26]. In this sense, it is worth noting

that rigid and amorphous (or "glassy" in Physics words) domains provide a thermodynamic advantage for the formation of short-lived interactions, since the free energy cost of their formation is low. Moreover, by including missing residues in soft disorder, we can capture two *disorder-to-order effects*. First, we can detect the total highly mobile regions that are sometimes missing and sometimes structured in alternative crystals (i.e. hot loops). Second, we can identify those regions that are missing in all alternative complexes (in a node of the hierarchy) and structured upon more complex oligomerisation. It is known that disorder-to-order regions are often involved in protein assembly [18, 22]. In summary, our measure of soft disorder identifies regions called *soft disordered regions* (SDRs) of the protein sequence, grouping residues that either have a high B-factor (see below) or are missing from at least one of the alternative crystals in the PDB that resolve the protein's interactions, see Fig 1A. In practice, only missing residues that undergo a transition from disorder to order in the protein cluster are used for the soft disorder-interface correlation analysis, since it is impossible to judge whether a missing residue belongs to an interface or not. This condition covers all residues that are intermittently ordered/disordered in different crystals with the same interaction complexity (typically highly mobile regions such as hot loops), or entire missing regions that undergo a disorder-to-order transition upon binding.

The use of B-factors for statistical studies has several complications. First, the B-factor is a measure of the error made in estimating the atomic coordinates, so its scale is determined mainly by the resolution of the experiment. In addition, B-factors are strongly affected by crystal defects and structural disorder, leading to problems of reproducibility between experiments [28, 29]. In this work, we are interested in precisely identifying the regions of the protein where the experiments fail. To do this, in order to get a complete picture, we need to combine information from alternative experiments where possible (we will discuss this process later when we consider the construction of hierarchies) and compare the results of different experiments and conformations. It is well known that a comparison between B-factors is only meaningful if they are normalised in the crystal [27–29]. This means that when calculating the SDRs, we are not interested in the absolute value that the B-factor reaches in a given experiment, but only in the atoms that have an anomalously high B-factor compared to the rest of the protein chain. In what follows, we will consider the B factor of a residue $i$, $B_i$, to be the B factor of its $C_\alpha$ atom. Then, to identify the flexible or floppy regions of a protein, we will rely on a normalised version of the B-factor, which we call *b-factor* (where b is written in lower case):

$$b_i = \frac{B_i - \langle B \rangle}{\sigma_B},$$ (1)

Where $\langle B \rangle$ and $\sigma_B$ are the mean and standard deviation of all $B_i$ in the protein chain (i.e. *B* is normalised in the chain, not in the protein complex). To define the SDR, we then need to set a static threshold for *b* to denote the difference between ordered and disordered AAs. The implications of this threshold have been discussed in detail in [25]. In particular, high thresholds, for example $b > 3$ (i.e. only those AAs with a B-factor greater than $3\sigma_B$ are considered), are more likely to form an interface in alternative structures of a given protein than regions with $b > 0.5$. However, since regions with $b > 3$ are much shorter than regions with $b > 0.5$, new interfaces are much more likely to be covered by the SDR defined with $b > 0.5$ than with $b > 3$. We have found that the best approximate trade-off between positive predictive value and sensitivity is achieved with a threshold $b > 1$.

For this reason, from now on, we will say that an AA is softly disordered if its $b_i > 1$ (which affects on average 16.7% of the AAs in a chain) or if its structure is missing in at least one of

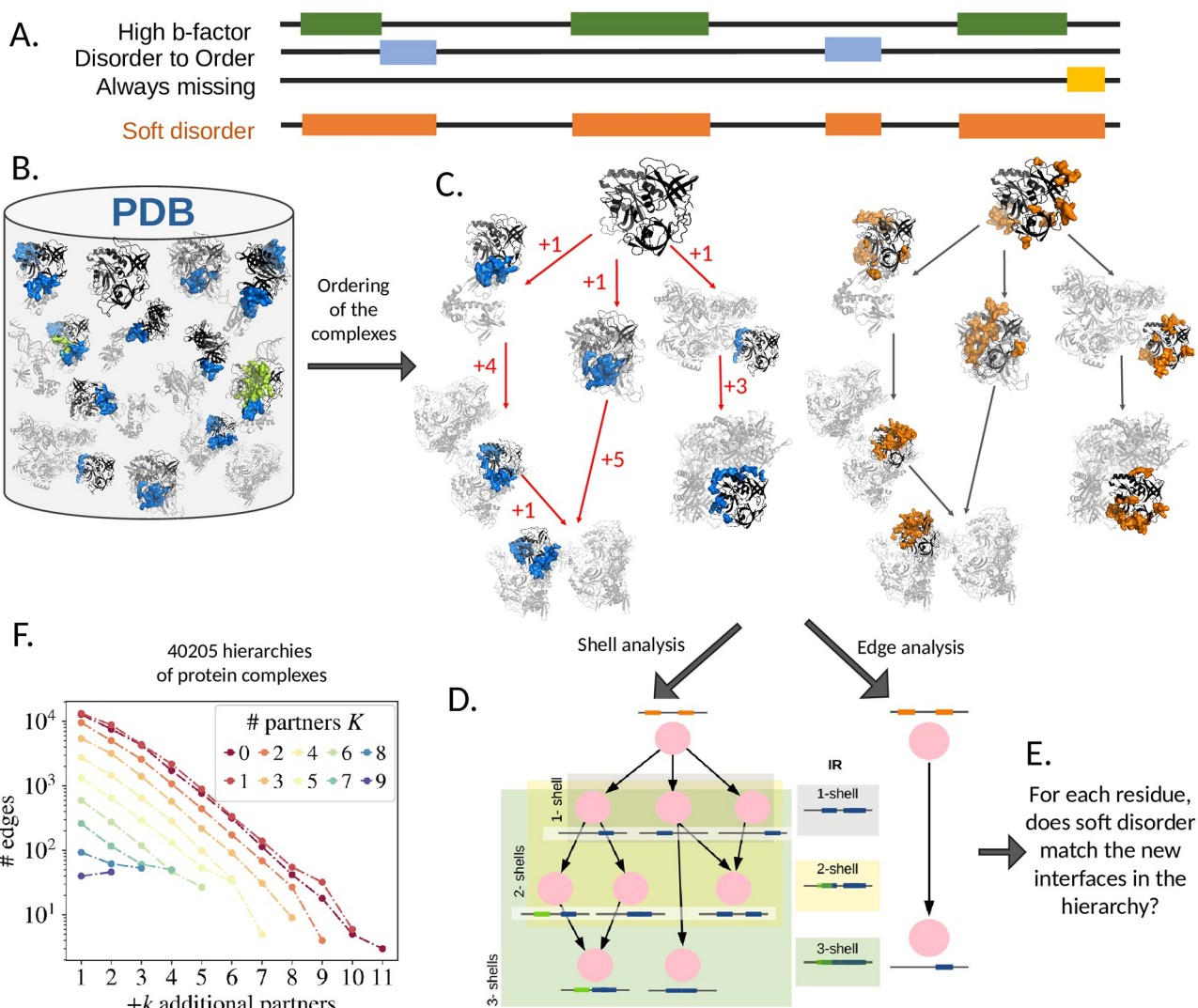

**Fig 1. A.** Scheme illustrating the definition of soft disorder for protein structures with the same interaction complexity, i.e. corresponding to a node in the hierarchy of assembly consisting of one or more alternative PDB crystals. Softly disordered residues are selected either for having high B-factor or for being missing in at least one of the hierarchy node's crystals. This means that both intermittently missing/structured residues (e.g. hot loops) and always missing residues are included. **B-E.** Scheme illustrating the construction and analysis of a hierarchy of protein structures using a cluster of PDB structures. **B.** All PDB entries that have a specific or very similar chain sequence (black) are grouped in the same cluster. The partners are shown in grey and the interfaces in blue and green, depending on whether the interaction is with a protein or with DNA. **C.** The interactions of a protein with its partners (B) are used to construct a hierarchy of progressive assembly represented by a DAG (S1 and S2 Figs). The labels on the red edges show the number of new partners interacting with the chain in the child node compared to the parent node. On the right hand side, the soft disorder (orange) of each node is calculated as in A. **D**: Schematic representation of the two different tests of correlation between the soft disorder in a parent node (orange regions on a black sequence) and the interface regions in its descendants (blue or green regions on a black sequence): *edge analysis* compares two nodes directly connected by an edge in the graph, and *shell analysis* compares the soft disorder of a given node with the union of all the interfaces of the chain in the shell of its descendants. A *k-shell* denotes the union of all interfaces in nodes that have $\leq k$ partners more than the input node of the DAG. *all-shell* denotes the union of all interfaces in the chain within the entire hierarchy. The union of the interface regions in each shell is sketched on the right. Each $k$-shell contains all the interface residues of the $i$-shells, with $i \leq k$. **E**: Our analysis aims to decide whether the soft disorder predicts new interfaces or not. **F**: Distribution of the total number of edges for proteins in the 40 205 hierarchies in our database containing parent nodes with $K$ partners, where $K$ varies from 0 (red) to 9 (blue) and $K + k$ indicates the number of additional partners $+k$ in the child of the edge (x-axis).

the crystals defining a node of the hierarchy of progressive assembly (see below). Note that in [25] both types of residues were studied separately and the reasons are due to several facts reported in [25], where we showed statistically that: (i) the majority of the missing residues in the PDB were intermittently disordered/ordered residues when many crystals of the same

protein were available, (ii) these disordered/ordered residues were also independently classified as high b-factor residues in protein clusters with more than 10 crystals in almost 100% of the cases, (iii) such residues were unlikely to be identified as missing by bioinformatics predictors of intrinsic disorder, even though hot loops are known to be easily predicted [30], and finally (iv) the "always missing" residues (i.e., regions that are missing in all nodes of a hierarchy), excluded in our current analysis, were predicted very accurately from sequences. All these reasons justify considering missing residues that undergo transitions from disorder to order and high b-factor residues, as two manifestations of the same effect, and to distinguish them from the group of intrinsic disordered residues ("always missing") whose role in assembly is likely to be different and much more difficult to assess.

## Hierarchy of interactions in the PDB

To test the effect of soft disorder on the progressive assembly of a protein complex, we need to order all protein structures available in the PDB according to the degree of oligomerisation. In particular, we used all the information available in the Bank up to 7 January 2022 and selected those structures obtained by X-ray diffraction experiments, a total of 155749 structures. In practice, the first step in building our interaction hierarchies is to assemble all PDB structures (i.e. their PDBID) containing a given protein sequence, together with their identification within the complex (i.e. their chain name). In practice, we considered two slightly different sequences to be equivalent (and thus contained in the same protein cluster) if they were equal up to 90% of sequence identity for the 90% of their length. For details on the clustering procedure, see Materials and methods. In total, we analysed the *interaction complexity* of 51332 different clusters (40205 have more than 1 structure) of very similar protein sequences. By interaction complexity of a given cluster, we mean all multiple interactions involving the reference protein, i.e. all interactions with different partners (even if they share the same binding region) or with identical partners but at different sites of its structure, partially overlapping or completely separated.

As in [25], we denote each cluster with the PDBID and the chain name of one of the structures forming it. The protein chain that gives the name to the group is considered the *representative sequence* of the cluster and is used to map the information observed in alternative structures to the same sequence for the cluster. We show an example of cluster construction in the box in Fig 1B. Once the cluster is built, we group its structures into nodes with similar interface regions (IRs) and identical partners, and arrange these nodes along a *directed acyclic graph (DAG)* of progressive assembly, with new branches reflecting new partners added to the previous parent structure. See Fig 1C for an example of a few branches, and S1 Fig for an example of an entire graph construction. See the Materials and methods section for a detailed explanation of the graph construction. The complexity of the DAG describes what we have previously called the complexity of a protein's interactions. We note that these DAGs can have multiple input nodes (or root nodes), i.e. nodes that have no incoming edge, and that an input node can correspond to either a complex or an unbound structure. A node is a leaf of the DAG (i.e. it has no offspring) if there is no structure in the PDB with an increasing number of partners that contains its interaction complexity.

By construction, our DAG edges connect nodes with a variable number of partners $K$, the only constraint being that the number of partners of the descendants, say $K'$, is greater than $K$ and that the partners and interface region of the parent are included in the descendant nodes. This means that the edges do not necessarily add only one new partner (i.e. a "+1" in Fig 1C); such a complex could either be inherently unstable or simply never observed). For the following analysis, which we will discuss later, we find it useful to keep track of the *degree* of each

relationship in the DAG, i.e. the number of partners added by the child. We denote this degree by $k$. In Fig 1D we show the total number of edges connecting a parent node with $K$ partners to a child node with $K + k$ partners.

Once the hierarchy of interactions is built for each cluster of protein chains, we need to assign a IR and a SDR to each node of the graph (as the union of all IRs/SDRs in the structures contained in that node). To do this, we align the sequences of each structure to the representative structure of the cluster and label a residue as part of the IR or the SDR if it has been labelled as an IDR/SDR in at least one of the structures of the node. Interactions with proteins and DNA/RNA are treated as two different types of IRs. As an example, in S2 Fig we graphically show the IRs (in blue for protein interactions and in green for DNA interactions) and the SDRs (in red) at each node for the entire hierarchy discussed in Fig 1B.

## Comparison between the soft disorder in the parents and the location of the new interfaces in the offspring

Now that we have ordered all the information about IRs and SDRs along the hierarchies of interface complexity, we can test the hypothesis that soft disorder modulates the location of new interfaces during complex oligomerisation. In practice, we compare the location (residue by residue in the sequence) of the parent SDRs with the new IRs observed in the progeny (by "new IR" we mean the IR residues that were not already labelled as IR in the parent. The missing residues that are missing in the offspring nodes are removed from the analysis because we cannot know if they belong to the interface or not. This means that IDRs that remain unstructured in all nodes of the DAG are never counted for the analysis. We can compare both measures at each level of the hierarchies with different standard tests such as sensitivity, specificity, accuracy, positive predictive value (PPV) and negative predictive value (NPV). We show the definition of these measures and the expectation for purely random correlations in the Materials and methods section.

We quantify the interface predictive power of the SDR of a parent node in two different ways: either we compare it with the new IRs found in a given direct descendant (*edge analysis*) or with the union of the new IRs observed in the descendants (*shell analysis*; see Fig 1C). In the latter case, we ~~can~~ compute the union ~~only~~ up to a fixed number of new additional partners + $k$, and call it *k-shell analysis*, or we extend it to the union of all descendants and call it *all*-shell analysis. Shell tests are about exploring the propensity of SDR to form interfaces, rather than predicting particular IRs.

In Fig 2A, we show the sensitivity versus 1-specificity for all our $k$-shell predictions for parent nodes with at least two levels of offspring. In a purely random coincidence, all points in this test would follow the diagonal. We have colored the points in the figure according to their local density to highlight populated regions. We find that the 75% of our predictions are better than pure chance, although most of them are only slightly better than chance. We also emphasise that this result is still very meaningful, as our knowledge of all possible interactions of a given protein in the PDB is still extremely incomplete (hence, most missed correspondences between SDRs and new IRs must be counted as random). We show the equivalent PPV versus NPV curve in S3 Fig.

It is important to emphasise that the total size of the new IRs increases when the number of partners of the offspring differs from that of the parent, as does the size of the non-IR regions when more and more partners are considered. For this reason, it is important to analyse separately the quality of the prediction as a function of the number of partners $K$ of the parent and the number of new partners + $k$ of the offspring nodes. Hence, from now on, we will average our tests over parents with equal $K$ and offspring with equal + $k$. For the *k-shell* test, this means

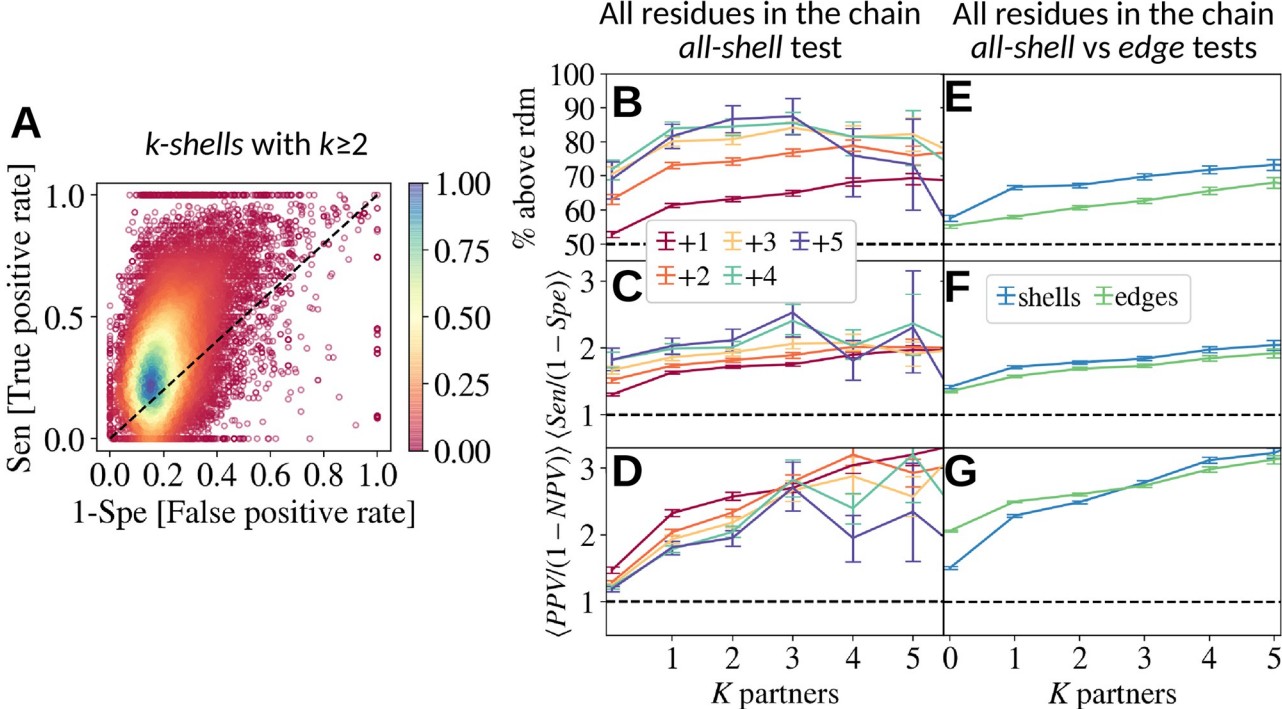

**Fig 2. A.** Analysis of all the predictions of the IRs evaluated in *k-shell* tests (for $k \leq 2$) for the entire set of clusters in our database. Each point in the plot reports sensitivity (Sen) versus 1-specificity (Spe) and corresponds to a prediction for a parent node. The color encodes the local density of that region. Random guesses should lie on the diagonal line. In this case, 75% of the predictions lie above this line. **B.** Percentage of predictions with Sen > 1-Spe conditioned to shells with at least $+k$ new partners (in different colours) and parents with $K$ neighbors. **C,D.** Same analysis as in B but for the ratios Sen/(1-Spe) and PPV/(1-NPV), respectively. **E, F, G.** Comparison of the *all-shell* statistics from previous panels with the analogous statistics extracted from the tests on the *edges*. In all cases, the horizontal line marks the random expectation.

offspring with $\leq k$ new partners (as explained in Fig 1D). In Fig 2B we show the percentage of predictions that are better than pure chance as a function of $K$ (different colours refer to different *k*-shells). In Fig 2C and 2D we show the ratios of sensitivity/(1-specificity) and PPV/(1-NPV). In all three figures, the random expectation is shown as a horizontal black dotted line. In all averaging groups, the predictions are better than random and improve with $+k$, supporting the idea that SDR encourages or enables the uptake of new interfaces in that area, without a strict choice of where and with whom. We also see that the predictive power improves with increasing the oligomerisation degree $K$ of the parent, which is mainly related to the fact that the size of the available surface decreases with $+k$. We will discuss this effect later and try to eliminate it. In Fig 2E, 2F and 2G we compare the *all-shell* statistics with those obtained with the *edge* analysis (recall Fig 1D). Again, we see that both tests give a statistically meaningful correlation between SDRs and new IRs, even though *shell* predictions seem to be slightly better than single interface predictions.

We further examine the quality of the predictions in Fig 3 via the averaged sensitivity, specificity, accuracy, PPV and NPV measures. As before, we average the test data by groups of predictions of equal parent's $K$ or equal number $+k$ of offspring. We also consider the "all-$K$" and "all+$k$" situations if all $K$ or $+k$ tests, respectively, are averaged together. In Fig 3A and 3B, we show data for the *shell* test and in Fig 3C and 3D, that for the *edge* test. The results of our tests

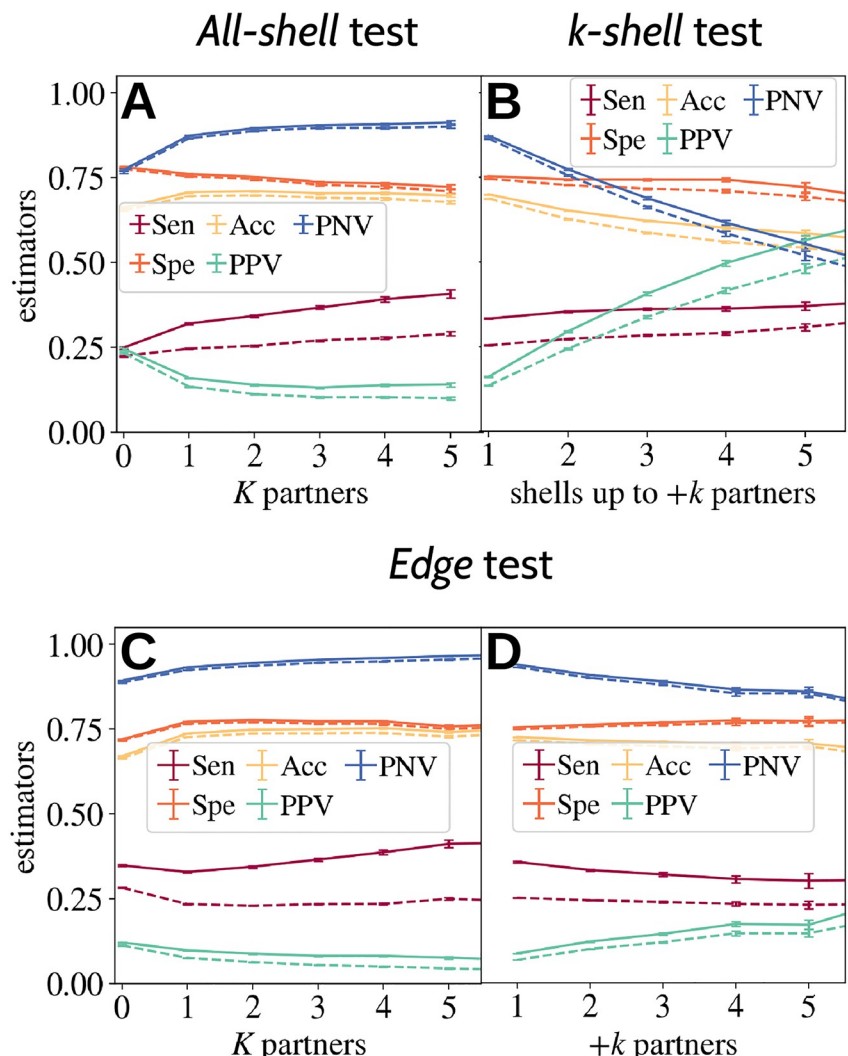

**Fig 3. Average sensitivity (Sen), specificity (Spe), accuracy (Acc), positive predictive value (PPV) and negative predictive value (NPV) of predictions based on knowledge of soft disorder in a parent. A.** The *all-shell* prediction quality is evaluated as a function of *K*, the number of partners of the parent node. **B.** The *k-shell* predictions is evaluated as a function of *k*, the maximum depth (in terms of new partners) of the shell used to compute the union of the interfaces. **C, D.** The goodness of the prediction of the *edge test* is evaluated either as a function of the parent *K* or as a function of +*k*, the number of new partners, of the direct offspring. In all tests, solid lines refer to the actual test and dotted lines to the random guess test.

are shown in solid lines, while the dotted lines are obtained when averaging the random expectation for each prediction, as discussed in the Materials and Methods Section.

While the effect of *K* is rather limited in both tests, the effect of +*k* is very strong in the *k-shell*-test (see Fig 3B), where both the real and the random PPV increase sharply with increasing *k*. This is nothing but a direct consequence of the fact that the whole new interface region grows with the addition of partners, so it becomes easier to predict it correctly by chance. However, we can see that the distance between the real curve and the random curve is mostly constant, which tells us that the intrinsic predictive power of a given SDR increases only mildly with the depth of the interaction shells considered (i.e. +*k*). This effect is much smaller in the *edge* test, where +*k* only marks the difference between the number of partners of nodes

connected by an edge (IRs from different nodes are not combined in this test). It is worth noting that the typical size of the predicted IR is extremely similar in both tests (about 30–40% of the total new IR), while PPV (and the significant quotient $\overline{\mathrm{PPV}}/\overline{\mathrm{PPV^r}}$) is higher in the shell tests. This combination supports the idea that an important part of the new IRs are accommodated in regions that were SDR in the ancestor, but also that a particular ancestor carries information about the new interfaces in the progeny. In both types of analysis, we find that the predictions for unbound nodes ($K = 0$) are significantly worse than the other predictions. This effect seems to be related to the existence of quite different SDRs in different unbound structures (which was illustrated with an example in [25]). These very different SDRs could be precursors of certain individual branches of the graph. An statistical analysis of this effect would require examining different nodes associated with unbound structures, and the selection of preferred branches for statistical analysis. Such an analysis is beyond the scope of this article.

### Eliminating biases: Results on the protein surface

Finally, as mentioned several times before, the size of the protein surface available for new interfaces is expected to decrease as we go down the DAG, and thus the size of the new IR. In parallel, IR residues tend to have lower b-factors while residues with high b-factors tend to be located at the surface of the protein. This means that we need to ensure that the correlation between SDR and new interfaces is not just the result of a reduction in the surface area available for new interactions and a stiffening of the "older" interaction regions. With this in mind, we consider three further tests to compare the data and the random expectations for the predictions for each node. Namely, instead of using the entire protein chain for the prediction as done up to now (the "C all" test), (i) we exclude from the analysis the residues belonging to the IR of the parent node (namely "C-IR", (ii) we consider only residues at the surface of the parent complex (namely "S all"), or (iii) we consider only residues at the surface that are not IR in the parent (the "S-IR" test). We stress that the total protein surface area is calculated in the entire protein complex of the parent node used to perform the test, specifically using the complex in the first (if many) of the PDBIDs contained in each node.

It is clear that (ii) and (iii) are very similar, since interfaces tend to be grounded. We show in Fig 4 the statistics of Fig 2 for these new tests. In Fig 4A, we compare the percentage of *all-shell* prediction points above the diagonal as a function of the number of partners of the parent node for the 4 tests (i.e. the original test with the whole protein chain plus the 3 new tests). In Fig 4B and 4C, we show the averaged value of the quotients Sen/(1-Spe) and PPV/(1-NPV), respectively, also as a function of $K$. We find that while the predictions are generally worse than those calculated with the whole chain, there is still an important correlation between SDR and new IRs, and all tests follow very similar trends. In Fig 4D, 4E and 4F, we show the dependence of $K$ in the most restricted test (which uses only the S-IR residues) and we observe that the main difference from Fig 2D and 2F is the disappearance of the strong dependence on $K$ and $+k$, as expected according to previous reasoning. We emphasise that this last test has fewer statistics than the original test because not all IR residues are on the protein surface, which forced us to withdraw many hierarchical relationships in our graphs.

### Soft disorder and AF2 low confidence regions

Very low values of the AF2 confidence metric pLDDT has been reported to correlate well with intrinsic disorder [2]. We have observed that this connection is much stronger and extends to SDR and IRs, with some nuances. As done for our notion of SDR, we label as "low-confidence" regions (LCR) the residues with pLDDT below the protein's backbone average. In Fig 5B, we consider two examples of crystal structures and their associated hierarchy of complexes in the

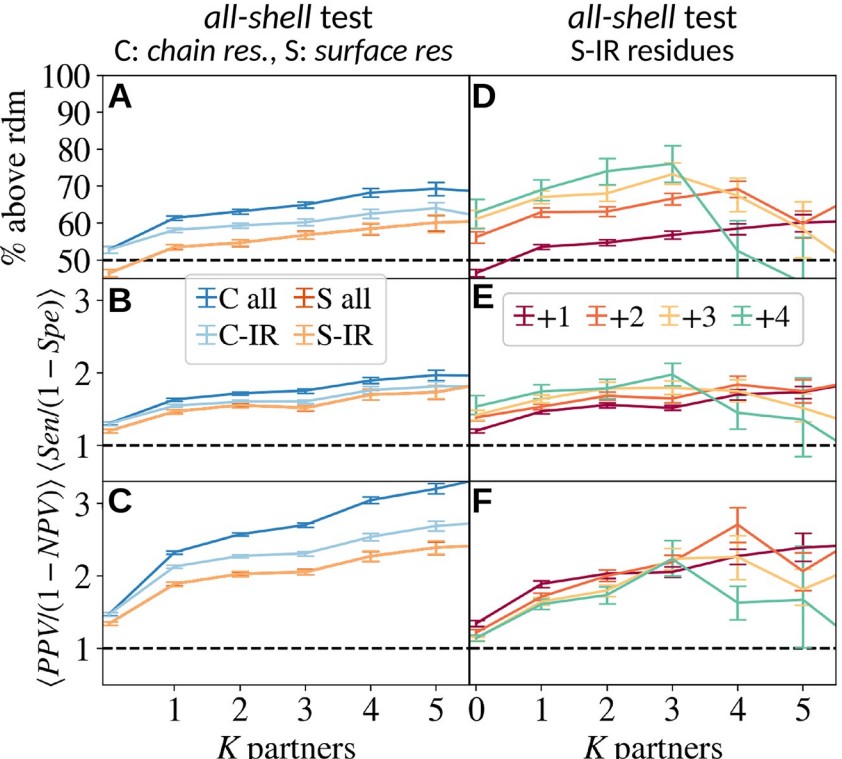

**Fig 4. We compare the statistics of [Fig 2] after removing trivial correlations between SDR and new IRs, associated to both kinds of regions preferring the protein surface and avoiding old interface regions.** We consider 4 different tests for the comparisons between SDR and new IRs: (C all) we use all the chain residues, (C-IR) we remove the IR residues in the ancestor from the test, (S all) we consider only the residues on the surface of the parent protein structure that remain accessible in the complex to which it belongs, and (S-IR), we use only those surface residues that did not belong to the IR in the ancestor. For all these tests, we show the percentage of *all*-shell prediction points having Sen > 1-Spe in **A** and the average of the quotients Sen/1-Spe in **B** and PPV/(1-NPV) in **C**, all them computed after grouping together predictions from nodes with the same number of partners $K$. In **D, E, F**, we study the dependence of the shells $K$ degree in these statistics for the most restricted test, the "S-IR".

PDB. For these two examples, we predicted the AF2 model structures of the unbound form and a complexified conformation. LCR in AF2 structural models of the unbound form correspond well to SDR measured on crystal structures over the entire hierarchy of interactions. When the complexified AF2 model is considered, one observes that its LCR remains essentially unchanged compared to the LCR computed in the unbound form, in contrast to SDR measured on the corresponding crystal structures. Then, we compare AF2 LCR measured on the unbound structure with the union of all the SDRs measured either in the protein hierarchy or in the unbound structure only in Fig 5A. Clearly, the relative pLDDT seems to correlate well with the SDR computed over the hierarchy, in contrast to the SDR computed on the unbound structure. Most importantly, when we compare the spatial location of AF2 LCR with the union of all the IRs present in the hierarchy, we observe an excellent correspondence in all the clusters tested. This means that the relative uncertainty in the structure predictions of AF2 carries information about the interaction network of a given chain, even if it remains unclear how to use it to predict a particular assembly path yet. To conclude, we want to stress that our LCRs may exist in model structures that are predicted with high confidence by AF2, that is with a global pLDDT > 80 or 90 over all residues. Finally, we emphasise that this entire last

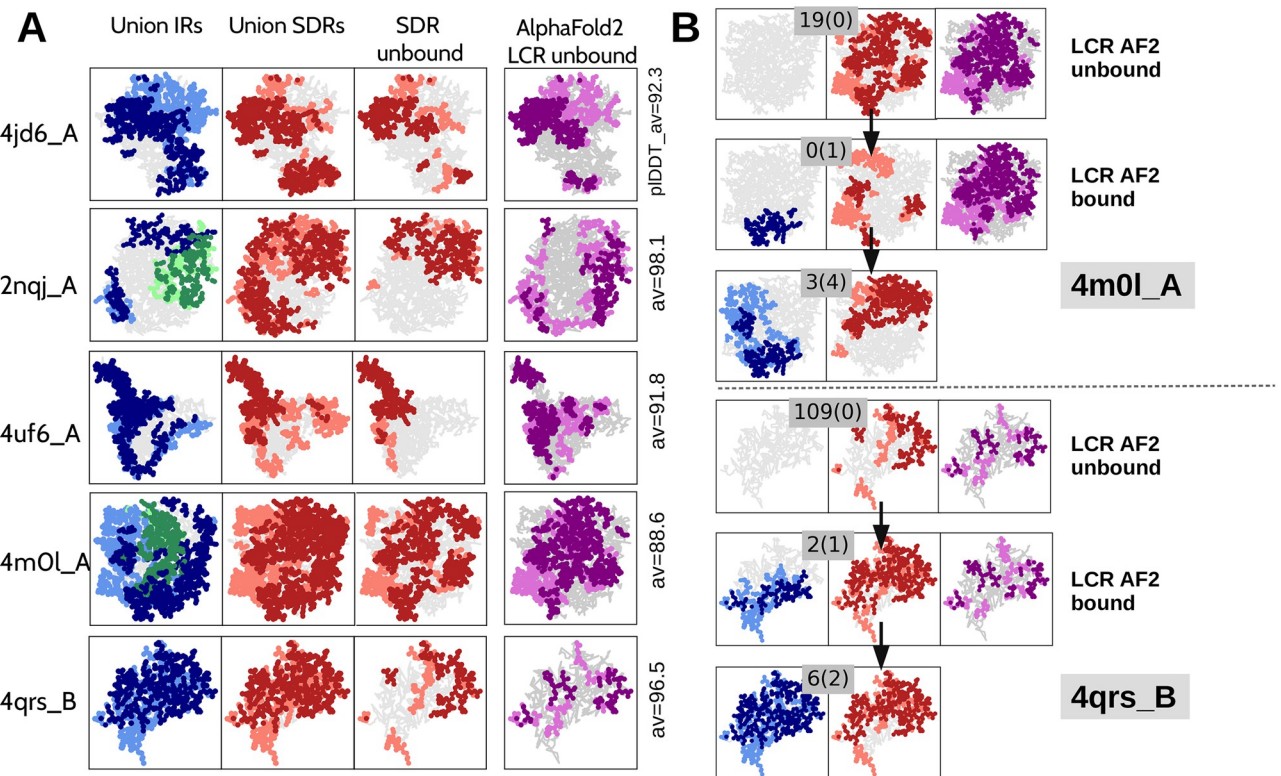

**Fig 5. IR and SDR analysis of five protein clusters (4jd6_A, 2nqj_A, 4uf6_A, 4m0l_A and 4qrs_B) and comparison with AF2 uncertainty (i.e. low confident regions, LCR). A** The union of all interface regions (IR, blue for proteins and green for DNA and RNA; first column) and the union of soft disorder regions (SDR, red; second column) identified by graph analysis are compared with the regions of the AF2 model that predict with high uncertainty the structure of the unbound protein (i.e. the pLDDT is below the protein average; purple; fourth column). For each chain, the average pLDDT value for all residues is shown on the right. The SDR value calculated for the unbound chain is also shown (third column). The structures and regions have been projected in two dimensions, and residues that are forward (with respect to the centre) are shown in a darker shade than those that are backward. **B.** Two hierarchies, for 4m0l_A and 4qrs_B, are described. The AF2 low confidence region (LCR) is calculated for the unbound and bound forms, resulting in the same levels of uncertainty, while the corresponding SDR changes from the unbound to the bound form.

discussion about AF2 LCRs is based on the study of a few specific examples and should be tested statistically in future work.

## Discussion

We have shown that there is a significant correlation between the position occupied by soft disorder residues (flexible, missing or rigid but amorphous) in a protein complex and the location of new interfaces as more and more partners are added to the complex. We note that this correlation is rather limited when we do not have much information about the hierarchy of interactions, but becomes larger when more and more partners are known. These results are supported by a large-scale analysis of all structures in the PDB that occur in hierarchies where similar protein chains interact with an increasing number of partners. These hierarchies can be viewed as an incomplete proxy for the pathway of complex assembly. The picture that emerges from their analysis is that a large proportion of the new interface regions lie on the more floppy or amorphous parts of the surface of the simpler complex.

In this context, previous work has highlighted that flexibility is used in nature to tune between protein conformations and functions in monomers [24]. Then one can hypothesise that dimer formation might also regulate function. From our results, this seems indeed a much

broader mechanism of functional emergence and regulation. In parallel, the existence of a disorder-based directional and sequential mechanism for the assembly of complexes was previously proposed in the context of intrinsically disordered proteins under the name of the bond-switching binding change model [22]. Our work provides statistical evidence for the generality of this phenomenon and for the convenience of treating flexibility and intrinsic disorder on an equal footing in the context of protein assembly. Regarding the role of intrinsic disorder, however, it is important to stress that our analysis formally considers only those missing residues that undergo a transition from disorder to order at some point in the hierarchy, i.e. either highly mobile residues that can be temporarily perceived as disordered or ordered, e.g. hot loops, or entire missing regions that become structured upon binding. There are probably two different types of intrinsic disorder, one that is directly related to binding, and another that always remains disordered. This effect was quantified in our earlier work [25], where we observed that disorder-to-order residues tended to be overwhelmingly identified as high b-factor residues whenever there was a large knowledge of alternative structures for a given protein.

With regard to the role of intrinsic disorder, however, it is important to emphasise that our analysis formally considers only the missing residues that undergo a transition from disorder to order. In this sense, only two types of missing residues are included in the measure of soft disorder: the missing residues that are missing only in some of the protein structures of a node (structures with the same interaction complexity), e.g. hot loops, and residues that are missing in the parent structure but ordered in the progeny, i.e. regions that undergo a disorder-to-order transition. The residues that are missing in both the parent and the descendant structure are therefore ignored in our analysis, as there is no way to judge whether they belong to the interface of the progeny or not. It is unlikely that such regions are directly involved in the construction of large complexes, as they are never structured. In our previous work [25], we have highlighted that these "always missing" regions are easily predicted from the sequence using predictors of intrinsic disorder, which means that they can be easily inferred and excluded for practical applications related to the assembly of complexes.

Determining the next assembly step during the formation of a complex governed by soft regions has a number of direct implications for the design of computational strategies aimed at reconstructing the full path of assembly: If the partner is available, one can identify where it will bind; if no partner is available, information about soft disordered regions helps to greatly reduce the number of potential interactors by restricting the search to specific areas of the surface. This is particularly useful when the set of potential partners has been experimentally identified but their interactions are still unknown.

Knowledge of soft disorder for protein-structure pairs could be used systematically in protein-docking experiments to greatly reduce the conformational search space, as was done when considering predicted interfaces before [31, 32].

Exciting hypotheses can be made about how knowledge of soft disordered regions can be crossed with predictions from AF2. In [2, 10] it was observed that the regions where intrinsic disorder is present correspond to those with high uncertainty for AF2. In this work, we have shown several examples suggesting that regions with relatively low pLDDT correspond to the union of all regions with soft disorder in our protein interaction hierarchies. AF2 can be used to reduce the search space in predicting 3D complexes to specific regions of the protein surface and specific partners. If the AF2 signal on unbound forms could be disentangled to predict the new/next? disordered regions after binding, then the low confidence AF2 regions identified on unbound forms could be useful to define appropriate strategies for sorting the next interacting region during assembly. Our results suggest that tuning pLDDT scores to track assembly can help overcome shortcomings associated with experimental B-factor determinations [27–29].

## Materials and methods

### Error bars

Error bars of Figs [1], [2], [3] and [4] represent the 90% confidence interval, and are computed using the Bootstrap method with 500 re-samplings.

### Clustering procedure

Clusters of a particular or very similar protein chains in the PDB are created using the MMSEQS2 method [33, 34]. Different protein sequences are clustered only if their sequence identity is greater than or equal to 90% and if their chain length is equal up to 90%. The components of each cluster are then a list of PDBID entries with a chain name that identify the protein structure in the crystal. The representative chain of the hierarchy gives its name and is selected as the PDBID of the higher resolution or R-value experiment.

### Interface computation

The protein-protein binding residues for each PDB crystal included in the analysis are calculated using the INTERFACE BUILDER method [35] (two AAs are considered to be in contact as long as their two $C_\alpha$ are within $\leq 5\text{Å}$ of each other).

To obtain the protein-DNA/RNA binding sites, we look for the residues whose relative surface area (RASA) decreases after binding. The change in RASA is calculated with NACCESS [36] (with a probe size of 1.4Å).

### Alignment to the representative chain

The IR and the SDR residues extracted from each protein structure included in a hierarchy are mapped to the representative chain of the protein cluster via sequence alignment with the Biopython's [37] PAIRWISE2.ALIGN.GLOBALXX routine.

### Construction of the hierarchy

For each protein chain structure in the cluster, we record the identity of all partners (same protein, different proteins, DNA or RNA) and the binding residues that connect the cluster protein to its partners. To determine whether our protein interacts with the same partner in different crystals (by same partner we mean proteins with identical or very similar sequences), we label the partner chains with the name of the cluster in which they are contained (i.e. the representative chain of that cluster). At this point, we use this information (position of the IRs in the sequence and the identity of the partners) to build a *directed acyclic graph (DAG)* of increasing complexity of the interactions of a given protein chain. This graph serves as a proxy for the hierarchy of the progressive assembly of that protein; see the graph in Fig 1C or S2 Fig. We start by grouping all chains that contain very similar IRs and exactly the same partners (see details above). Each of these different groups represents a node of the graph. We will say that a node has $K$ partners if our protein has binding sites with $K$ chains. We emphasise that the chains of these partners can be either similar chains, another protein, DNA or RNA. After we have created all the nodes (see details below), we add directed edges (or arrows) to the graph showing the increase in partners and IRs.

All nodes with $K' > K$ connected to a node with $K$ partners are called *offspring* of this common *parent* node. In our construction, two nodes can be connected by an edge (i.e. to be "related") only if the descendant contains all partners of the parent and new additional ones, and if more than 75% of IR observed in the parent is also present in the descendant. Moreover, two nodes satisfying these conditions are connected by an edge only if no third complex

structure in the cluster can be placed between them in the genealogical reconstruction. An example of this construction can be seen in Fig 1B. A node is a root of the DAG if it has no parent. When the root is also an unbound node (i.e. when $K = 0$), it is a common ancestor of all structures in the cluster. However, this is not necessarily the case for clusters without unbound structures, where the DAG may consist of several root nodes or several disconnected graphs with possibly an unknown common ancestor.

All structures with the same *interaction complexity* are grouped into one node, and the IR and SDR of the node are considered as the union of all IR and SDR residues measured in each of the node components. We assumed that two or more protein structures in the cluster have the same interaction complexity if two conditions are met: (i) the number and identity of their partners are identical (i.e. they have the same $K$, and the cluster identity of all their partners is identical) and (ii) the intersection between all IRs of the structures in the node is greater than the 75% of each of these IRs. For this analysis, protein and DNA interfaces are treated as completely different types of interfaces, which means that condition (ii) must be satisfied separately for each type of interface. All DNA/RNA partners are considered to be the same partner.

## Goodness of the prediction tests

If the same residue is marked as SDR in an ancestor and as new IR in a descendant, we say that the prediction is a true positive (TP). However, if the residue is not a new IR in the progeny, we say it is a false positive (FP). Conversely, residues that are labelled as new IRs in the progeny but not in the ancestor's SDR are false negatives (FN) and true negatives (TN) if the residue is neither a new IR nor an SDR in the two related nodes. We can combine these numbers to obtain different estimators for the *goodness* of the prediction:

$$\text{Sensitivity(Sen)} = \frac{\text{TP}}{\text{TP} + \text{FN}}, \tag{2}$$

$$\text{Specificity(Spe)} = \frac{\text{TN}}{\text{FP} + \text{TN}}, \tag{3}$$

$$\text{Accuracy(Acc)} = \frac{\text{TP} + \text{TN}}{\text{TP} + \text{FP} + \text{TN} + \text{FN}}, \tag{4}$$

$$\text{Precision or Positive Predictive Value(PPV)} = \frac{\text{TP}}{\text{TP} + \text{FP}}, \tag{5}$$

$$\text{Negative Predictive Value(NPV)} = \frac{\text{TN}}{\text{TN} + \text{FN}}. \tag{6}$$

Sen quantifies the proportion of new IR, correctly predicted by the SDR of the ancestor, and Spe, the same but for those non-IR. Acc indicates the proportion of the total residues whose role in the progeny was correctly predicted. Finally, PPV indicates the proportion of IR predictions that are actually new IR in the progeny, and NPV indicates the proportion of non IR predictions that are actually not IR in the progeny.

In a chain containing $L$ residues, a totally random prediction of $N_\text{D}$ SDR residues, would predict correctly (in average) a new IR residue with probability $r_\text{I} = N_\text{I}/L$, where $N_\text{I}$ is the number of new IR residues in the descendant. Similarly, in a random guess scenario, one expects the following values for the above estimators: $\langle \text{Sen}^\text{r} \rangle = r_\text{D}$, $\langle \text{Spe}^\text{r} \rangle = 1 - r_\text{D}$, $\langle \text{Acc}^\text{r} \rangle = r_D(2r_I - 1)$

$+ (1 - r_I) \langle PPV^r \rangle = r_I, \langle NPV^r \rangle = 1 - r_I$, with $\langle r_D \rangle = N_D/L$). In other words, $\langle Sen^r \rangle = 1 - \langle Spe^r \rangle$ and $\langle PPV^r \rangle = 1 - \langle NPV^r \rangle$.

## AF2 structure predictions and pLDDT

AF2 and AF2-multimer have proven to be able to accurately predict the 3D structure of individual proteins or protein complexes based on amino acid sequence. Each residue in a model structure is accompanied by a measure of its reliability on a scale of 0 to 100, based on the pLDDT metric. Regions with pLDDT>90 are thus modelled with very high accuracy, while regions with pLDDT<50 are classified as uninterpreted and are known to be strongly correlated with the presence of intrinsic disorder [2]. In order to compare the pLDDT values with our definition of soft disorder (which is normalised over chains), we did not focus on absolute values, but on values that are relative to the mean over the chain. Indeed, the purple regions shown in Fig 5 are defined by residues with confidence values below the mean. We call these regions *low confidence regions* or LCR for short. In practise, we used the ColabFold [38] web server to obtain the AF2 predictions and calculate the average pLDDT in the protein chain considering the 5 different AF2 models provided. The list of regions with the lowest confidence results from the union of all residues with a pLDDT lower than the average of the five models.

## Supporting information

**S1 Fig. Details of the PDBID and chain index of all components of the interaction hierarchy of protein cluster 4m0l_A (the same cluster discussed in Fig 1).** The blue circles group all chain structures contained in this node. Red edges highlight the "genealogical" relationships shown in Fig 1C.
(TIF)

**S2 Fig. The complexity of the interactions and the associated soft disorder for the protein sequence associated with cluster 4m0l_A, discussed in Fig 1.** To improve visualisation, we have projected the 3-dimensional structures into a 2-dimensional sketch. Interface regions are shown in blue or green depending on whether the interaction is with another protein or with DNA, regions of soft disorder are coloured red, and residues that are forward are shown in a darker shade than those that are backward. Each node of the graph is labelled by two numbers "$N(M)$", where $N$ is the identifier of the node and $M$ if the number of partners of the protein at node $N$. Note the input node of the graph, labelled 19(0), where $M = 0$ indicates the unbound form. **A.** An equivalence between the 3-dimensional representation of the protein and the sketch in 2-dimensions for the structure of node $N = 0$. In the 3D structures, the interfacial residues are shown in blue (top left) and the soft residues in orange (top right). **B.** The whole hierarchy of interactions from the PDB. Horizontal lines ($K$p.) summarise all protein complexes with a fixed number of partners $K$. The arrows indicate the increase in the number of partners from top to bottom. With respect to the unbound structure in node 19(0) (blue square), all interactions contained in line 1p. define the 1-shell of interactions in Fig 1D. In the same way, all the interfaces in lines 1p. and 2p. define the 2-shell of interactions, and so on. The union of all interfaces in this graph forms the *all*-shell. If another predecessor node is considered as the origin, such as node 12(2) (orange square), its interaction shells would consist only of the nodes with a higher $K$ connected to it. That is, node 11 for the 1-shell and nodes 9 and 11 for the 2-shell.
(TIF)

**S3 Fig. We show the PPV vs. 1-NPV for all *k-shell* tests (with $k \leq 2$) predictions in our database.** Each point corresponds to a prediction for a parent node. The colour encodes the local density of this region. As in Fig 2A, the 75% of the predictions lie above the random guess line.
(TIF)

## Author Contributions

**Conceptualization:** Beatriz Seoane, Alessandra Carbone.

**Data curation:** Beatriz Seoane.

**Formal analysis:** Beatriz Seoane.

**Funding acquisition:** Beatriz Seoane, Alessandra Carbone.

**Investigation:** Beatriz Seoane, Alessandra Carbone.

**Methodology:** Beatriz Seoane, Alessandra Carbone.

**Project administration:** Beatriz Seoane, Alessandra Carbone.

**Software:** Beatriz Seoane.

**Supervision:** Alessandra Carbone.

**Validation:** Beatriz Seoane.

**Visualization:** Beatriz Seoane.

**Writing – original draft:** Beatriz Seoane, Alessandra Carbone.

**Writing – review & editing:** Beatriz Seoane, Alessandra Carbone.

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
