## [Decision Letter · Decision Letter 0]

4 Aug 2022

Dear Dr. Seoane,

Thank you very much for submitting your manuscript "Soft disorder modulates the assembly path of protein complexes" for consideration at PLOS Computational Biology. As with all papers reviewed by the journal, your manuscript was reviewed by members of the editorial board and by several independent reviewers. The reviewers appreciated the attention to an important topic. Based on the reviews, we are likely to accept this manuscript for publication, providing that you modify the manuscript according to the review recommendations.

Please make sure to also address the concerns from Reviewer 1: The topic is interesting, and the conclusions potentially interesting. The only concern I have is that the IDR based interactions often happen with high on/off rates, and I am not sure to what extent they are involved in building up large complexes. The authors might want to comment on this in the discussion.

Sincerely,

Rune Linding

Guest Editor

PLOS Computational Biology

Arne Elofsson

Deputy Editor

PLOS Computational Biology

[LINK]

Please make sure to also address the concerns from Reviewer 1: The topic is interesting, and the conclusions potentially interesting. The only concern I have is that the IDR based interactions often happen with high on/off rates, and I am not sure to what extent they are involved in building up large complexes. The authors might want to comment on this in the discussion.

Reviewer's Responses to Questions

**Comments to the Authors:**

Reviewer #1: The topic is interesting, and the conclusions potentially interesting. The only concern I have is that the IDR based interactions often happen with high on/off rates, and I am not sure to what extent they are involved in building up large complexes. The authors might want to comment on this in the discussion.

Reviewer #2: In this manuscript the authors study the role of protein flexibility in protein complex assembly. The work presented here follows from a previous study from the same authors (Seoane et al. PLOS Comp Bio 2021) where they have shown some degree of association between protein flexibility , estimated by a normalised B-factor, and interface residues. In the previous publication, these authors hinted at the possibility that flexibility could be predictive of interface regions along a path of assembly from monomeric to larger assemblies. This suggestion is now followed up statistically in the current manuscript. Essentially, the single point of the manuscript is that interface residues of complexes having larger numbers of subunits can be somewhat predicted by the flexibility of residues in complexes having lower number of subunits. This result is shown in figure 5 with figures 6,7 and 8 essentially trying to show that the result is robust to different ways of looking at the data. More broadly the relation between interface residues and flexibility has a very long history of study with quite many perspectives, including ideas around folding upon binding; flexibility of core vs. periphery of interface residues; and changes in entropy upon binding among other ideas and literature. The work done here, where the authors study changes in flexibility in a given protein with increasing numbers of bound interactors is a useful addition to the literature.

I only have a small number of concerns and they are primarily related with presentation and clarity instead of technical issues:

1 - The biggest issue I have had while reviewing the paper was the clarity of the manuscript itself. There is essentially one main result in the paper but there are 9 figures with the fist real result starting in figure 5. Figures 1 to 4 are diagrams and a non critical result figure. Figures 6, 7 and 8 are controls and figure 9 is a very short result related with Alphafold predictions that probably should be removed. The text itself is also very difficult to follow because the authors introduce many acronyms and symbols (SDR, IR,k, K, PPV, etc) and some paragraphs are really full with acronyms and sentences one needs to re-read a few times to understand. It would be great if the authors could simplify and improve on the readability of the work.

1.1 – I suggest the authors could simplify and merge figures 1, 2 and 3 and move figure 4 to supplementary.

1.2 – I don’t really see the point of adding the Alphafold result at the end of the manuscript. It is a small example and as the authors state, it would require a proper analysis. I don’t see how it adds to the central claim of the work. It would make more sense to develop into a proper stand alone piece of work. Instead, the authors could leave some of these ideas in the discussion section alone without any specific results.

1.3 – There are some parts of the results that could be moved to the methods section. Page 5 and 6 – the definitions of how the DAG was built could be, at least in part, moved to the methods. Page 9, around lines 202-206, the definition of random might be better in the methods.

1.4 – It is hard to say what to do with the acronyms but, at least in places, the authors could try to avoid using them to make it easier for readers that are not as familiar with their terms.

2 – One point that stood out from a technical perspective was that the authors have analysed the results in the form of how often the TPR and FPR are better than random. One of the key result is in Figure 5A where the authors report that 74% of the predictions (i.e. flexible regions predicting the next interface region) are better than random. While this seems to be the case I think it is also noteworthy that for the majority of these cases the predictions are really just somewhat better than random. For a given FPR most of the predictions have a TPR that is quite close but above random. Overall, this association reported in the manuscript seems real but having a small effect size. I think it would be important to comment on this in the discussion section.

3 – The authors make a big point, both here and in the previous manuscript about the term “soft disorder”. I don’t really see the need to come up with a new term, why not just use the term flexibility ? Or relative flexibility since the authors used a protein normalized metric.

Additional minor suggestions:

- It is not common to underline words in scientific communications, might be better to remove this

- The abstract usually opens with some form of introductory sentence(s) as to why the work/topic is important.

**Have the authors made all data and (if applicable) computational code underlying the findings in their manuscript fully available?**

Reviewer #1: None

Reviewer #2: Yes

PLOS authors have the option to publish the peer review history of their article (what does this mean?). If published, this will include your full peer review and any attached files.

Reviewer #1: No

Reviewer #2: No

Figure Files:

Data Requirements:

Reproducibility:

References:

---

## [Decision Letter · Decision Letter 1]

1 Nov 2022

Dear Dr. Seoane,

Thank you very much for submitting your manuscript "Soft disorder modulates the assembly path of protein complexes" for consideration at PLOS Computational Biology. As with all papers reviewed by the journal, your manuscript was reviewed by members of the editorial board and by several independent reviewers. The reviewers appreciated the attention to an important topic. Based on the reviews, we are likely to accept this manuscript for publication, providing that you modify the manuscript according to the review recommendations.

Please fix the repeated paragraphs

Sincerely,

Arne Elofsson

Section Editor

PLOS Computational Biology

Arne Elofsson

Section Editor

PLOS Computational Biology

Please fix the repeated paragraphs

Reviewer's Responses to Questions

**Comments to the Authors: **

Reviewer #1: The manuscript has been improved and the main message is now easier to get. Thank you.

Reviewer #2: The authors have addressed my previous concerns, in particular by making the manuscript more accessible to others and by adding a small note in the discussion about the small effect size of the effect they are describing here. Not exactly those words but at least it is mentioned. 

In the PDF version I read there is a repeated paragraph in line 203 starting with "By construction .."

**Have the authors made all data and (if applicable) computational code underlying the findings in their manuscript fully available?**

Reviewer #1: None

Reviewer #2: Yes

PLOS authors have the option to publish the peer review history of their article (what does this mean?). If published, this will include your full peer review and any attached files.

Reviewer #1: No

Reviewer #2: No

Figure Files:

Data Requirements:

Reproducibility:

References:

---

## [Editor Report · Decision Letter 2]

6 Nov 2022

Dear Dr. Seoane,

We are pleased to inform you that your manuscript 'Soft disorder modulates the assembly path of protein complexes' has been provisionally accepted for publication in PLOS Computational Biology.

Best regards,

Arne Elofsson

Section Editor

PLOS Computational Biology

Arne Elofsson

Section Editor

PLOS Computational Biology

---

## [Editor Report · Acceptance letter]

11 Nov 2022

PCOMPBIOL-D-22-00788R2 

Soft disorder modulates the assembly path of protein complexes

Dear Dr Seoane,

I am pleased to inform you that your manuscript has been formally accepted for publication in PLOS Computational Biology. Your manuscript is now with our production department and you will be notified of the publication date in due course.

With kind regards,

Anita Estes
